# S-TLLR: STDP-inspired Temporal Local Learning Rule for Spiking Neural Networks

**Marco P. E. Apolinario**                                                    *mapolina@purdue.edu*
*School of Electrical and Computer Engineering*
*Purdue University*

**Kaushik Roy**                                                               *kaushik@purdue.edu*
*School of Electrical and Computer Engineering*
*Purdue University*

**Reviewed on OpenReview:** *https://openreview.net/forum?id=CNaiJRcX84*

## Abstract

Spiking Neural Networks (SNNs) are biologically plausible models that have been identified as potentially apt for deploying energy-efficient intelligence at the edge, particularly for sequential learning tasks. However, training of SNNs poses significant challenges due to the necessity for precise temporal and spatial credit assignment. Back-propagation through time (BPTT) algorithm, whilst the most widely used method for addressing these issues, incurs high computational cost due to its temporal dependency. In this work, we propose S-TLLR, a novel three-factor temporal local learning rule inspired by the Spike-Timing Dependent Plasticity (STDP) mechanism, aimed at training deep SNNs on event-based learning tasks. Furthermore, S-TLLR is designed to have low memory and time complexities, which are independent of the number of time steps, rendering it suitable for online learning on low-power edge devices. To demonstrate the scalability of our proposed method, we have conducted extensive evaluations on event-based datasets spanning a wide range of applications, such as image and gesture recognition, audio classification, and optical flow estimation. S-TLLR achieves comparable accuracy to BPTT (within $\pm 2\%$ for most tasks), while reducing memory usage by $5 - 50\times$ and multiply-accumulate (MAC) operations by $1.3 - 6.6\times$, particularly when updates are restricted to the last few time-steps. [1]

## 1 Introduction

Over the past decade, the field of artificial intelligence has undergone a remarkable transformation, driven by a prevalent trend of continuously increasing the size and complexity of neural network models. While this approach has yielded remarkable advancements in various cognitive tasks (Brown et al., 2020; Dosovitskiy et al., 2021), it has come at a significant cost: AI systems now demand substantial energy and computational resources. This inherent drawback becomes increasingly apparent when comparing the energy efficiency of current AI systems with the remarkable efficiency exhibited by the human brain (Roy et al., 2019; Gerstner et al., 2014; Christensen et al., 2022; Eshraghian et al., 2023). Motivated by this observation, the research community has shown a growing interest in brain-inspired computing. The idea behind this approach is to mimic key features of biological neurons, such as spike-based communication, sparsity, and spatio-temporal processing.

Bio-plausible Spiking Neural Network (SNN) models have emerged as a promising avenue in this direction. SNNs have already demonstrated their ability to achieve competitive performance compared to more traditional Artificial Neural Networks (ANNs) while significantly reducing energy consumption per inference when deployed in the right hardware (Davies et al., 2018; Sengupta et al., 2019; Christensen et al., 2022;

---

[1]Code available at https://github.com/mapolinario94/S-TLLR

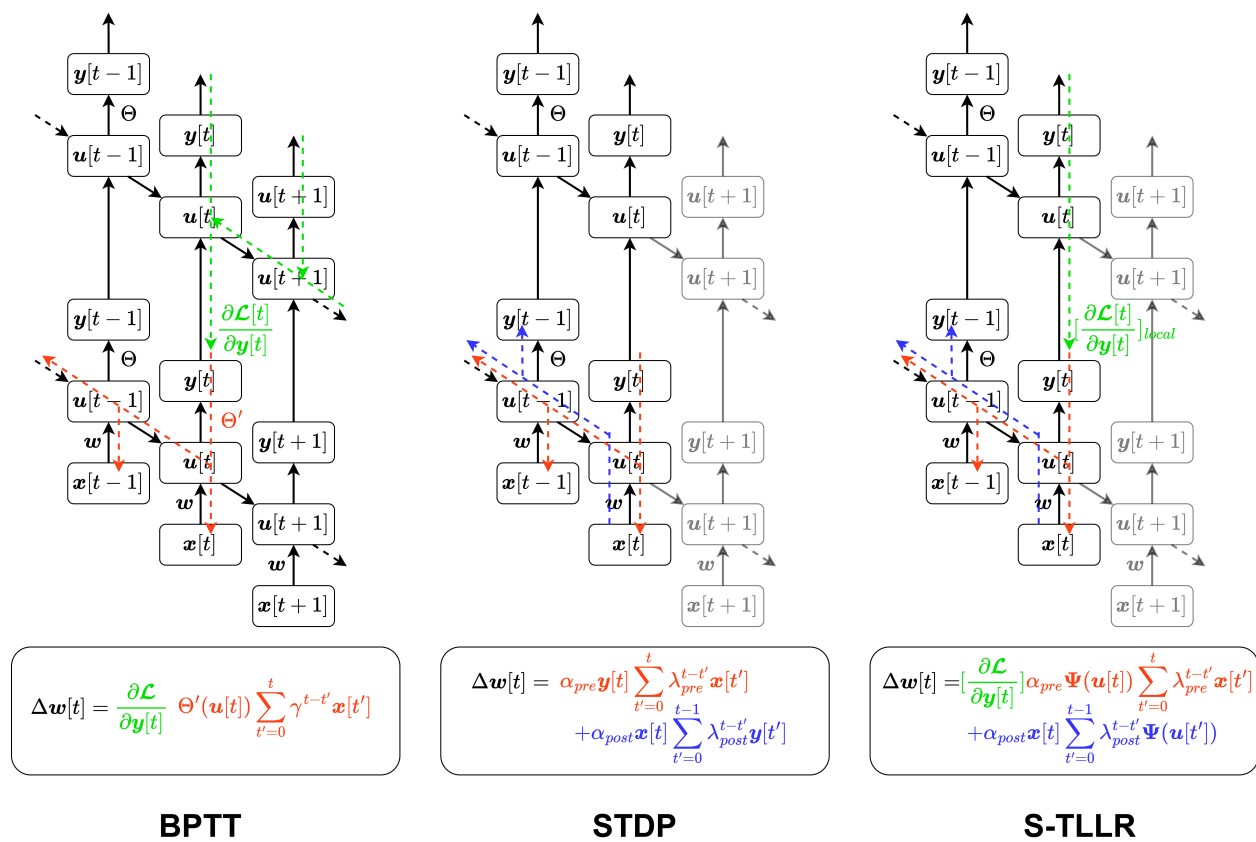

Figure 1: Comparison of weight update computation of a feed-forward spiking layer for BPTT, STDP, and our proposed learning rule S-TLLR. The spiking layer is unrolled over time for the three algorithms while showing the signals involved in the weight updates. The top-down learning signal is shown in green, while the signals locally available to the layer are represented as red for the causal term and blue for non-causal terms. Also, note that the learning signal in BPTT relies on future time steps, whereas in S-TLLR, this signal is computed locally in time.

Neftci et al., 2019; Roy et al., 2019). One of the main advantages of SNNs lies in their event-driven binary sparse computation and temporal processing based on membrane potential integration. These features make SNNs well-suited for deploying energy-efficient intelligence at the edge, particularly for sequential learning tasks (Ponghiran & Roy, 2022; Bellec et al., 2020; Christensen et al., 2022).

Despite their promise, training SNNs remains challenging due to the necessity of solving precisely temporal and spatial credit assignment problems. While traditional gradient-based learning algorithms, such as back-propagation through time (BPTT), are highly effective, they incur a high computational cost (Eshraghian et al., 2023; Bellec et al., 2020; Bohnstingl et al., 2022). Specifically, BPTT has memory and time complexity that scales linearly with the number of time steps ($T$), such as $O(Tn)$ and $O(Tn^2)$, respectively, where $n$ is the number of neurons, making it unsuitable for edge systems where memory and energy budgets are limited. This has motivated several studies that propose learning rules with approximate BPTT gradient computation with constant memory (Bellec et al., 2020; Bohnstingl et al., 2022; Quintana et al., 2023; Ortner et al., 2023; Xiao et al., 2022). However, most of those learning rules have a complexity that scales with the number of synapses ($O(n^2)$), as shown in Table 1, making them expensive for deep convolutional SNN models in practical scenarios (where $n \gg T$). Moreover, as most of those learning rules have been derived from BPTT, they only leverage causal relations between the timing of pre- and post-synaptic activities, over-looking non-causal relations used in other learning mechanisms, such as Spike-Timing Dependent Plasticity (STDP) (Bi & Poo, 1998; Song et al., 2000).

Table 1: Comparison of S-TLLR with learning methods from the literature (Where $n$ is the number of neurons and $T$ total number of time steps) [Note that complexities are expressed in big $O$ notation for one single layer]

| Method | Memory Complexity | Time Complexity | Temporal Local | Leverage Non-Causality |
|---|---|---|---|---|
| BPTT | $Tn$ | $Tn^2$ | X | X |
| RTRL (Williams & Zipser, 1989) | $n^3$ | $n^4$ | ✓ | X |
| e-prop (Bellec et al., 2020) | $n^2$ | $n^2$ | ✓ | X |
| OSTL (Bohnstingl et al., 2022) | $n^2$ | $n^2$ | ✓ | X |
| ETLP (Quintana et al., 2023) | $n^2$ | $n^2$ | ✓ | X |
| OSTTP (Ortner et al., 2023) | $n^2$ | $n^2$ | ✓ | X |
| OTTT (Xiao et al., 2022) | $n$ | $n^2$ | ✓ | X |
| **S-TLLR (Ours)** | $n$ | $n^2$ | ✓ | ✓ |

To overcome the above limitations, in this paper, we propose S-TLLR, a novel three-factor temporal local learning rule inspired by the STDP mechanism. Specifically, S-TLLR is designed to train SNNs on event-based learning tasks while incorporating both causal and non-causal relationships between the timing of pre- and post-synaptic activities for updating the synaptic strengths. This feature is inspired by the STDP mechanism from which we propose a generalized parametric STDP equation that uses a secondary activation function to compute the post-synaptic activity. Then, we take this equation to compute an instantaneous eligibility trace (Gerstner et al., 2018) per synapse modulated by a third factor in the form of a learning signal obtained from the backpropagation of errors through the layers (BP) or by using fixed random feedback connections directly from the output layer to each hidden layer (Trondheim, 2016). Notably, S-TLLR exhibits constant (in time) memory and time complexity, making it well-suited for online learning on resource-constrained devices.

In addition to this, we demonstrate through experimentation that including non-causal information in the learning process results in improved generalization and task performance. Also, we explored S-TLLR in the context of several event-based tasks with different amounts of spatio-temporal information, such as image and gesture recognition, audio classification, and optical flow estimation. For all such tasks, S-TLLR can achieve performance comparable to BPTT and other learning rules, with significantly lower memory and computation requirements.

The main contributions of this work can be summarized as:

- We introduce a novel temporal local learning rule, S-TLLR, for spiking neural networks, drawing inspiration from the STDP mechanism, while ensuring a memory complexity that scales linearly with the number of neurons and remains constant over time.

- Demonstrate through experimentation the benefits of considering non-causal relationships in the learning process of spiking neural networks, leading to improved generalization and task performance.

- Validate the effectiveness of the proposed learning rule across a diverse range of network topologies, including VGG, ResNet, U-Net-like, and recurrent architectures.

- Investigate the applicability of S-TLLR in various event-based camera applications, such as image and gesture recognition, audio classification, and optical flow estimation, broadening the scope of its potential uses.

## 2 Background

### 2.1 Spiking Neural Networks (SNNs)

To model the neuronal dynamics of biological neurons, we use the leaky integrate and fire (LIF). The LIF model can be mathematically represented as follows:

$$u_i[t] = \gamma(u_i[t-1] - v_{\text{th}}y_i[t-1]) + w_{ij}x_j[t] \tag{1}$$

$$y_i[t] = \Theta(u_i[t] - v_{\text{th}}) \tag{2}$$

where $u_i$ represents the membrane potential of the $i$-th neuron, and $w_{ij}$ is the forward synaptic strength between the $i$-th post-synaptic neuron and the $j$-th pre-synaptic neuron. Additionally, $\gamma$ is the leak factor that reduces the membrane potential over time, $v_{\text{th}}$ is the threshold voltage, and $\Theta$ is the Heaviside function. Thus, when $u_i$ reaches the $v_{\text{th}}$, the neuron produces an output binary spike $(y_i)$. This output spike triggers the reset mechanism, represented by the reset signal $v_{\text{th}}y_i[t]$, which reduces the magnitude of $u_i$.

## 2.2 Spike-Timing Dependent Plasticity (STDP)

STDP is a learning mechanism observed in various neural systems, from invertebrates to mammals, and is believed to play a critical role in the formation and modification of neural connections in the brain in processes such as learning and memory (Gerstner et al., 2014). STDP describes how the synaptic strength $(w_{ij})$ between two neurons can change based on the temporal order of their spiking activity. Specifically, STDP describes the phenomenon by which $w_{ij}$ is potentiated if the pre-synaptic neuron fires just before the post-synaptic neuron fires, and $w_{ij}$ is depressed if the pre-synaptic neuron fires just after the post-synaptic neuron fires. This means that STDP rewards causality and punishes non-causality. However, Anisimova et al. (2022) suggests that STDP favoring causality can be a transitory effect, and over time, STDP evolves to reward both causal and non-causal relations in favor of synchrony. Such general dynamics of STDP can be described by the following equation:

$$\Phi(t_i, t_j) = \begin{cases} \alpha_{pre}\lambda_{pre}^{t_i-t_j} & \text{if } t_i \geq t_j \text{ (causal term)} \\ \alpha_{post}\lambda_{post}^{t_j-t_i} & \text{if } t_i < t_j \text{ (non-causal term)} \end{cases} \tag{3}$$

where $\Phi(t_i, t_j)$ represents the magnitude of the change in the synaptic strength $(\Delta w_{ij})$, $t_i$ and $t_j$ represent the firing times of the post- and pre-synaptic neurons, $\alpha_{pre}$ and $\lambda_{pre}$ are strength and exponential decay factor of the causal term, respectively. Similarly, $\alpha_{post}$ and $\lambda_{post}$ parameterize the non-causal effect. Note that when $\alpha_{post} < 0$, STDP favors causality, whereas $\alpha_{post} > 0$, STDP favors synchrony.

Based on STDP, the change in the synaptic strengths at time $t$ can be computed forward in time using local variables by the following learning rule (Gerstner et al., 2014):

$$\Delta w_{ij}[t] = \alpha_{pre}y_i[t]\sum_{t'=0}^{t}\lambda_{pre}^{t-t'}x_j[t'] + \alpha_{post}x_j[t]\sum_{t'=0}^{t-1}\lambda_{post}^{t-t'}y_i[t'] \tag{4}$$

$$w_{ij}[t+1] = w_{ij}[t] + \Delta w_{ij}[t] \tag{5}$$

where $y_i[t]$ and $x_j[t]$ are binary values representing the existence of post- and pre-synaptic activity (spikes) at time $t$, respectively. Note that both summations in (4) can be computed forward in time as a recurrent equation of the form $\text{tr}(x_j)[t] = \lambda_{\text{pre}}\text{tr}(x_j)[t-1] + x_j[t]$, where $\text{tr}(x_j)[t]$ is a trace of $x_j$. Hence, (4) can be expressed as:

$$\Delta w_{ij}[t] = \alpha_{pre}y_i[t]\text{tr}(x_j)[t] + \alpha_{post}x_j[t](\text{tr}(y_i)[t] - y_i[t]) \tag{6}$$

## 2.3 BPTT and three-factor (3F) learning rules

BPTT is the algorithm by default used to train spiking neural networks (SNNs) as it is able to solve spatial and temporal credit assignment problems. BPTT calculates the gradients by unfolding all layers of the network in time and applying the chain rule to compute the gradient as:

$$\frac{d\mathcal{L}}{d\boldsymbol{w}} = \sum_{t}^{T}\frac{\partial\mathcal{L}}{\partial\boldsymbol{y}[t]}\frac{\partial\boldsymbol{y}[t]}{\partial\boldsymbol{u}[t]}\frac{\partial\boldsymbol{u}[t]}{\partial\boldsymbol{w}} \tag{7}$$

Although BPTT can yield satisfactory outcomes, its computational requirements scale with time, posing a limitation. Moreover, it is widely acknowledged that BPTT is not a biologically plausible method, as highlighted in Lillicrap & Santoro (2019).

In contrast, 3F learning rules (Gerstner et al., 2018) are a more biologically plausible method that uses the combination of inputs, outputs, and a top-down learning signal to compute the synaptic plasticity. The general idea of the 3F rules is based on that synapses are updated only if a signal called eligibility trace $e_{ij}$ is present. This eligibility trace is computed based on general functions of the pre- and post-synaptic activity decaying over time. Such behavior is modeled in a general sense on the following recurrent equation:

$$e_{ij}[t] = \beta e_{ij}[t-1] + f(y_i[t])g(x_j[t]) \tag{8}$$

Here $\beta$ is an exponential decay factor, $f(y_i[t])$ and $g(x_j[t])$ are element-wise functions of the post- and pre-synaptic activity, respectivetly. Then, the change of the synaptic strengths ($w_{ij}$) is obtained by modulating $e_{ij}$ with a top-down learning signal ($\delta_i$) as:

$$\Delta w_{ij} = \sum_t \delta_i[t]e_{ij}[t] \tag{9}$$

3F learning rules have demonstrated their effectiveness in training SNNs, as shown by Bellec et al. (2020). Additionally, it is possible to approximate BPTT using a 3F rule when the learning signal ($\delta_i[t]$) is computed as $\frac{\partial \mathcal{L}[t]}{\partial y[t]}$ (instantaneous error learning signal), and the eligibility trace approximates $\frac{\partial y[t]}{\partial u[t]}\frac{\partial u[t]}{\partial w}$, as discussed by Bellec et al. (2020) and Martín-Sánchez et al. (2022). However, it is important to note that 3F rules using eligibility traces formulated as (8) exhibit a memory complexity of $O(n^2)$, rendering them very expensive in terms of memory requirements for deep convolutional SNNs.

## 3 Related Work

### 3.1 Existing methods for training SNNs

Several approaches to train SNNs have been proposed in the literature. In this work, we focus on surrogate gradients(Neftci et al., 2019; Li et al., 2021), and bio-inspired learning rules (Diehl & Cook, 2015; Thiele et al., 2018; Kheradpisheh et al., 2018; Bellec et al., 2020).

Training SNNs based on surrogate gradient methods (Neftci et al., 2019; Li et al., 2021) extends the traditional backpropagation through time (BPTT) algorithm to the domain of SNNs, where the non-differentiable firing function is approximated by a continuous function during the backward pass to allow the propagation of errors. The advantage of these methods is that they can exploit the temporal information of individual spikes so that they can be applied to a broader range of problems than just image classification (Paredes-Vallés & de Croon, 2021; Cramer et al., 2022). Moreover, such methods can result in models with low latency for energy-efficient inference (Fang et al., 2021a). However, training SNNs based on surrogate gradients with BPTT incurs high computational and memory costs because BPTT's requirements scale linearly with the number of time steps. Hence, such methods can not be used for training online under the hardware constraints imposed by edge devices (Neftci et al., 2019).

Another interesting avenue is the use of bio-inspired learning methods based on the principles of synaptic plasticity observed in biological systems, such as STDP (Bi & Poo, 1998; Song et al., 2000) or eligibility traces (Gerstner et al., 2014; 2018), which strengthens or weakens the synaptic connections based on the relative timing of pre- and post-synaptic spikes optionally modulated by a top-down learning signal. STDP methods are attractive for on-device learning as they do not require any external supervision or error signal. However, they also have several limitations, such as the need for a large number of training examples and the difficulty of training deep networks or complex ML problems (Diehl & Cook, 2015). In contrast, three-factor learning rules using eligibility traces (neuron local synaptic activity) modulated by an error signal, like e-prop (Bellec et al., 2020), can produce more robust learning overcoming limitations of unsupervised methods such as STDP. Nevertheless, such methods' time and space complexity typically make them too costly to be used in deep SNNs (especially in deep convolutional SNNs).

### 3.2 Learning rules addressing the temporal dependency problem

As discussed in the previous section, the training methods based on surrogate gradient using BPTT results in high-performance models. However, their major limitations are associated with high computational requirements that are unsuitable for low-power devices. Such limitations come from the fact that BPTT has to store a copy of all the spiking activity to exploit the temporal dependency of the data during training. In order to address the temporal dependency problem, several methods have been proposed where the computational requirements are time-independent while achieving high performance. For instance, the Real-Time Recurrent Learning (RTRL) (Williams & Zipser, 1989) algorithm can compute exact gradients without the cost of storing all the intermediate states. Although it has not been originally proposed to be used on SNNs, it could be applied to them by combining with surrogate gradients (Neftci et al., 2019). More recently, other methods such as e-prop (Bellec et al., 2020), OSTL (Bohnstingl et al., 2022), and OTTT (Xiao et al., 2022), derived from BPTT, allows learning on SNNs using only temporally local information (information that can be computed forward in time). However, with the exception of OTTT, all of these methods have memory and time complexities of $O(n^2)$ or worse, as shown in Table 1, making them significantly more expensive than BPTT, when used for deep SNNs in practical scenarios (where $n \gg T$). Moreover, since those methods (with the exception of RTRL) have been derived as approximations of BPTT, they only use causal relations in the timing between pre- and post- synaptic activity, leaving non-causal relations (as those used in STDP shown in Fig. 1) unexplored.

### 3.3 Combining STDP and backpropagation

As previously discussed, STDP has been used to train SNNs models in an unsupervised manner (Diehl & Cook, 2015; Thiele et al., 2018; Kheradpisheh et al., 2018). However, such approaches suffer from severe drawbacks, such as requiring a high number of timesteps (latency), resulting in low accuracy performance and being unable to scale for deep SNNs. So, to overcome such limitations, there have been some previous efforts to use STDP in combination with backpropagation for training SNNs, by either using STDP followed for fine-tuning with BPTT (Lee et al., 2018) or modulating STDP with an error signal (Tavanaei & Maida, 2019; Hu et al., 2017; Hao et al., 2020). However, such methods either do not address the temporal dependency problem of BPTT or do not scale for deep SNNs or complex computer vision problems.

## 4 STDP-inspired Temporal Local Learning Rule (S-TLLR)

### 4.1 Overview of S-TLLR and its key features

We propose a novel 3F learning rule, S-TLLR, which is inspired by the STDP mechanism discussed in Section 2.2. S-TLLR is characterized by its temporally local nature, leveraging non-causal relations in the timing of spiking activity while maintaining a low memory complexity $O(n)$.

Regarding memory complexity, a conventional 3F learning rule requires an eligibility trace ($e_{ij}$) which involves a recurrent equation as described in (8). In such formulation, the $e_{ij}$ is a state requiring a memory that scales linearly with the number of synapses ($O(n^2)$). For the S-TLLR, we dropped the recurrent term and considered only the instantaneous term (i.e. $\beta = 0$ in (8)). Instead of requiring $O(n^2)$ memory to store the state of $e_{ij}$, we need to keep track of only two variables ($f(y_i)$ and $g(x_i)$) with $O(n)$ memory. Hence, $e_{ij}[t]$ can be computed as the right-hand side of (6) which exhibits a memory complexity $O(n)$. This low-memory complexity is a key aspect of S-TLLR since it enables the method to be used in deep neural models where methods such as Williams & Zipser (1989); Bellec et al. (2020); Bohnstingl et al. (2022); Ortner et al. (2023); Quintana et al. (2023) are considerably more resource-intensive.

Finally, since BPTT is based on the propagation of errors in time, it only uses causal relations to compute gradients, that is, the relation between an output $y[t]$ and previous inputs $x[t], x[t-1], \ldots, x[0]$. These causal relations are shown in Fig. 1 (red dotted line). Also, methods derived from BPTT (Bellec et al., 2020; Bohnstingl et al., 2022; Ortner et al., 2023; Quintana et al., 2023; Xiao et al., 2022) use exclusively causal relations. In contrast, we took inspiration from the STDP mechanisms, which use both causal and

non-causal relations in the spike-timing (Fig. 1, red and blue dotted lines), to formulate S-TLLR as a 3F learning rule with a learning signal modulating instantaneous eligibility trace signal as shown in Fig. 1.

## 4.2 Technical details and implementation of S-TLLR

As discussed in the previous section, our proposed method, S-TLLR has the form of a three-factor learning rule, $\Delta w_{ij}[t] = \delta_i[t]e_{ij}[t]$, involving a top-down learning signal $\delta_i[t]$ and an eligibility trace, $e_{ij}[t]$. To compute $e_{ij}[t]$, we use a generalized version of the STDP equation described in (4) that can use a secondary activation function to compute the postsynaptic activity.

$$e_{ij}[t] = \alpha_{pre}\Psi(u_i[t]) \sum_{t'=0}^{t} \lambda_{pre}^{t-t'} x_j[t'] + \alpha_{post}x_j[t] \sum_{t'=0}^{t-1} \lambda_{post}^{t-t'}\Psi(u_i[t']) \tag{10}$$

Here, $\Psi$ is a secondary activation function that can differ from the firing function used in (2). We found empirically that using a function $\Psi(u)$ with $\int \Psi(u)du \leq 1$ yields improved results, specific $\Psi$ functions are shown in Appendix A.3. Furthermore, (10) considers both causal (first term on the right side) and non-causal (second term) relations between the timing of post- and pre-synaptic activity, which are not captured in BPTT (or its approximations (Bellec et al., 2020; Bohnstingl et al., 2022; Xiao et al., 2022)). The causal (non-causal) relations are captured as the correlation in the timing between the current post- (pre-) synaptic activity and the low-pass filtered pre- (post-) synaptic activity. Note that (10) can be computed forward in time (expressing it in the form of (6)) and using only information locally available to the neuron as follows:

$$e_{ij}[t] = \alpha_{pre}\Psi(u_i[t])\mathrm{tr}(x_j)[t] + \alpha_{post}y_i[t](\mathrm{tr}(\Psi(u_i[t])) - \Psi(u_i[t]) \tag{11}$$

The learning signal, $\delta_i[t]$, is computed as the instantaneous error back-propagated from the top layer ($L$) to the layer ($l$), as shown in (12), and the synaptic update is done as shown in (13).

$$\delta_i^{(l)}[t] = \begin{cases} \frac{\partial \mathcal{L}(\boldsymbol{y}^L[t], \boldsymbol{y^*})}{\partial y_i^{(l)}[t]} & \text{if } t \geq T_l \\ 0 & \text{otherwise} \end{cases} \tag{12}$$

$$w_{ij} := w_{ij} + \rho \sum_{t=T_l}^{T} \delta_i[t]e_{ij}[t] \tag{13}$$

Here, $\rho$ is the learning rate, $T$ is the total number of time steps for the forward pass, $T_l$ is the initial time step for which the learning signal is available, $\boldsymbol{y^*}$ is the ground truth label vector, $\boldsymbol{y}^L[t]$ is the output vector of layer $L$ at time $t$, and $\mathcal{L}$ is the loss function. Note that the backpropagation occurs through the layers and not in time, so the S-TLLR is temporally local. Depending on the task, good performance can be achieved even if the learning signal is available just for the last time step ($T_l = T$).

While our primary focus is on error-backpropagation to generate the learning signal, it is worth noting that employing random feedback connections, such as direct feedback alignment (DFA) (Trondheim, 2016; Frenkel et al., 2021), for this purpose is also feasible. In such cases, S-TLLR also exhibits spatial locality. We present some experiments in this direction in Appedix C.1. The S-TLLR algorithm for a multilayer implementation is shown in Algorithm 1.

### 4.2.1 S-TLLR for models with recurrent synaptic connections

A more general model of the LIF model includes explicit recurrent connections. Similar to the equation (1), the LIF model featuring explicit recurrent connections can be mathematically represented as:

$$u_i[t] = \gamma(u_i[t-1] - v_{\mathrm{th}}y_i[t-1]) + w_{ij}^{\mathrm{ff}}x_j[t] + w_{ik}^{\mathrm{rec}}y_k[t-1] \tag{14}$$

$$y_i[t] = \Theta(u_i[t] - v_{\mathrm{th}}) \tag{15}$$

Here, $u_i$ denotes the membrane potential of the $i$-th neuron. Moreover, $w^{\text{ff}}ij$ represents the forward synaptic connection between the $i$-th post-synaptic neuron and the $j$-th pre-synaptic neuron, while $w^{\text{rec}}ik$ signifies the recurrent synaptic connection between the $i$-th and $k$-th neurons in the same layer. Other terms remain consistent with the description in (1).

The eligibility trace for the forward connections is computing following (10), however, since for the recurrent synaptic connections, the neuron outputs from a previous time step $(t - 1)$ serve as inputs for the current time step $(t)$ the eligibility trace change to encapsulated this behavior as follows:

$$e^{\text{rec}}_{ik}[t] = \alpha_{pre}\Psi(u_i[t])\sum_{t'=1}^{t}\lambda_{pre}^{t-t'}y_k[t'-1] + \alpha_{post}y_k[t-1]\sum_{t'=0}^{t-1}\lambda_{post}^{t-t'}\Psi(u_i[t']) \tag{16}$$

Note that for models with explicit recurrent connections, memory requirements remain constant, thereby upholding temporal locality.

### 4.3 Computational improvements

Here we analyzed the theoretical computational improvements in terms of the number of multiply-accumulate (MAC) operations and memory requirements of S-TLLR with respect to BPTT. First, we started by expanding the BPTT terms by replacing (1) on (7). Assuming an SNN model with $L$ layers and $N$ neurons per layer, and appropriately factorizing the terms as described in Appendix B.1, we obtain the following expression for the gradients on the synaptic connections in layer $l$:

$$\frac{d\mathcal{L}}{dw_{ij}^{(l)}} = \sum_{t'=0}^{T}\frac{\partial\mathcal{L}}{\partial y_j^{(l)}[t']}\Theta'(u_j^{(l)}[t'])\sum_{t=0}^{t'}\gamma^{t'-t}y_i^{(l-1)}[t] \tag{17}$$

From (34), it can be observed that BPTT is not local in time as the updated at time step $t'$ depend on future time steps, as illustrated in Fig. 1. Therefore, the BPTT requires information on all the time steps, and consequently, its memory requirements scale linearly with the total number of time steps in the input sequence $(T)$. Then, the total memory required is estimated as follows:

$$\text{Mem}_{\text{BPTT}} = T \times \sum_{l=0}^{L} N^{(l)} \tag{18}$$

To estimate the number of operations, specifically MAC operations, we exclude any element-wise operations. Referring to (34), we ascertain that the number of operations depends on both the number of inputs and outputs, resulting in a total of $N^{(l)} \times N^{(l-1)}$ operations. Additionally, we need to account for the operations involved in propagating the learning signals to the previous layer, equating to $N^{(l)} \times N^{(l-1)}$. Consequently, the estimated number of operations can be calculated as follows:

$$\text{MAC}_{\text{BPTT}} = 2T \times \sum_{l=1}^{L} N^{(l)} \times N^{(l-1)} \tag{19}$$

A similar analysis is done for S-TLLR. Based on the temporal locality of the method as expressed in (11), we can conclude that the memory requirements of S-TLLR can be computed as follows:

$$\text{Mem}_{\text{S-TLLR}} = 2 \times \sum_{l=0}^{L} N^{(l)} \tag{20}$$

Here, the factor 2 is produced by the trace variables required to maintain the temporal information. For the number MACs, we first must note that S-TLLR only updates the weights when the learning signal is present at time $T_l$, that is weights are updated $T - T_l$ times. Additionally, we account for the operations required by the non-causal terms. Therefore, the number of operations for S-TLLR is:

$$\text{MAC}_{\text{S-TLLR}} = 3(T - T_l) \times \sum_{l=1}^{L} N^{(l)} \times N^{(l-1)} \tag{21}$$

Based on the previous discussion, the theoretical improvements in memory ($S_{\text{mem}}$) and number of operations ($S_{\text{MAC}}$) can be obtained as follows:

$$S_{\text{mem}} = \frac{\text{Mem}_{\text{BPTT}}}{\text{Mem}_{\text{S-TLLR}}} = \frac{T \times \sum_{l=0}^{L} N^{(l)}}{2 \times \sum_{l=0}^{L} N^{(l)}} = \frac{T}{2} \tag{22}$$

$$S_{\text{MAC}} = \frac{\text{MAC}_{\text{BPTT}}}{\text{MAC}_{\text{S-TLLR}}} = \frac{2T \times \sum_{l=1}^{L} N^{(l)} \times N^{(l-1)}}{3(T - T_l) \times \sum_{l=1}^{L} N^{(l)} \times N^{(l-1)}} = \frac{2T}{3(T - T_l)} \tag{23}$$

It is important to note that these theoretical improvements do not account for the overhead introduced by specific hardware implementations, which may result in discrepancies between the theoretical and experimental results.

---

**Algorithm 1** S-TLLR algorithm

---

**Require:** $\boldsymbol{x}, \boldsymbol{y*}, \boldsymbol{w}, T, T_l, \rho$

1: **for** $t = 1, 2, \ldots, T$ **do**
2:     **for** $l = 1, 2, \ldots, L$ **do**
3:         Update membrane potential $\boldsymbol{u}^{(l)}[t]$ with (1)
4:         Produce output spikes $\boldsymbol{y}^{(l)}[t]$ with (2)
5:         Update eligibility trace $\boldsymbol{e}^{(l)}[t]$ with (11)
6:     **end for**
7:     **if** $t \geq T_l$ **then**
8:         Initialize $\boldsymbol{\delta}^L = \nabla_{\boldsymbol{y}^L[t]} \mathcal{L}(\boldsymbol{y}^L[t], \boldsymbol{y*})$
9:         Initialize $\boldsymbol{a}^{(L)} := \mathbf{1}$
10:        **for** $l = L - 1, L - 2, \ldots, 2$ **do**
11:           **if** learning signal produced by backpropagation **then**
12:             $\boldsymbol{\delta}^{(l)} = \boldsymbol{a}^{(l+1)} \odot (\boldsymbol{w}^{(l+1)\top} \boldsymbol{\delta}^{(l+1)})$
13:             $\boldsymbol{a}^{(l)} = \boldsymbol{\delta}^{(l)} \odot \Theta'(\boldsymbol{u}^{(l)}[t])$ {$\Theta'$ is a surrogate gradient}
14:           **else if** learning signal produced by random feedback **then**
15:             $\boldsymbol{\delta}^{(l)} = \boldsymbol{B}^{(l)} \boldsymbol{\delta}^{(L)}$ {$\boldsymbol{B}^{(l)}$ is a fixed random matrix}
16:           **end if**
17:        **end for**
18:        $\Delta w_{ij}^{(L)}[t] = \delta_i^L y_j^{(L-1)}[t]$ {Last layer}
19:        $\Delta w_{ij}^{(l)}[t] = \delta_i^{(l)} e_{ij}^{(l-1)}[t]$ {Hidden layers}
20:     **end if**
21: **end for**
22: **for** $l = 1, 2, \ldots, L$ **do**
23:     Update weights $\boldsymbol{w}^{(l)} := \boldsymbol{w}^{(l)} + \rho \sum_T \Delta \boldsymbol{w}^{(l)}[t]$
24: **end for**
25: **return** $\boldsymbol{w}$

---

## 5 Experimental Evaluation

### 5.1 Effects of non-causal terms on learning

We performed ablation studies on the DVS Gesture, DVS CIFAR10, N-CALTECH101, and SHD datasets to evaluate the effect of the non-causal factor ($\alpha_{post}$) on the learning process.

For this purpose, we train a VGG9 model, described in Appedix A, five times with the same random seeds for 30, 30, and 300 epochs in the DVS Gesture, N-CALTECH101, and DVS CIFAR10 datasets, respectively. Similarly, a recurrent SNN (RSNN) model, described in Appedix A, was trained five times during 200 epochs on the SHD. To analyze the effect of the non-causal term, we evaluate three values of $\alpha_{post}$, $-1$, 0, and 1. According to (10), when $\alpha_{post} = 0$, only causal terms are considered, while $\alpha_{post} = 1$ ($\alpha_{post} = -1$) means that the non-causal term is added positively (negatively). As shown in Table 2, for vision tasks, it can be

Table 2: Effects of including the non-causal terms in the eligibility traces during learning [Accuracy (mean±std) reported over 5 trials ]

| Dataset | Model | T | $T_l$ | $(\lambda_{\mathbf{post}}, \lambda_{\mathbf{pre}}, \alpha_{\mathbf{pre}})$ | $\alpha_{\mathbf{post}} = \mathbf{0}$ | $\alpha_{\mathbf{post}} = +\mathbf{1}$ | $\alpha_{\mathbf{post}} = -\mathbf{1}$ |
|---|---|---|---|---|---|---|---|
| DVS Gesture | VGG9 | 20 | 15 | (0.2, 0.75, 1) | $94.61 \pm 0.73\%$ | $94.01 \pm 1.10\%$ | $\mathbf{95.07 \pm 0.48}\%$ |
| DVS CIFAR10 | VGG9 | 10 | 5 | (0.2, 0.5, 1) | $72.93 \pm 0.94\%$ | $73.42 \pm 0.50\%$ | $\mathbf{73.93 \pm 0.62}\%$ |
| N-CALTECH101 | VGG9 | 10 | 5 | (0.2, 0.5, 1) | $62.24 \pm 1.22\%$ | $53.42 \pm 1.50\%$ | $\mathbf{66.33 \pm 0.86}\%$ |
| SHD | RSNN | 100 | 10 | (0.5, 1, 1) | $77.09 \pm 0.33\%$ | $\mathbf{78.23 \pm 1.84}\%$ | $74.69 \pm 0.47\%$ |

seen that using $\alpha_{post} = -1$ improves the average accuracy performance of the model with respect to only using causal terms $\alpha_{post} = 0$. In contrast, for SHD, using $\alpha_{post} = 1$ improves the average performance over using only causal terms, as shown in Table 2. This indicates that considering the non-causal relations of the spiking activity (either positively or negatively) in the learning rule helps to improve the network performance. An explanation for this effect is that the non-causal term acts as a regularization term that allows better exploration of the weights space. Additional ablation studies are presented in Appedix C.3 that support the improvements due to the non-causal terms.

## 5.2 Performance comparison

### 5.2.1 Image and Gesture Recognition

We train VGG-9 and ResNet18 models during 300 epochs using the Adam optimizer with a learning rate of 0.001. The models were trained five times with different random seeds. The baseline was set using BPTT, while the models trained using S-TLLR used the following STDP parameters $(\lambda_{post}, \lambda_{pre}, \alpha_{post}, \alpha_{pre})$: $(0.2, 0.75, -1, 1)$ for DVS Gesture and $(0.2, 0.5, -1, 1)$ for DVS CIFAR10 and N-CALTECH101.

The test accuracies are shown in Table 3. In all those tasks, S-TLLR shows a competitive performance compared to the BPTT baseline. In fact, for DVS Gesture and N-CALTECH101, S-TLLR outperforms the average accuracy obtained by the baseline trained with BPTT. Because of the small size of the DVS Gesture dataset and the complexity of the BPTT algorithm, the model overfits quickly resulting in lower performance. In contrast, S-TLLR avoids such overfitting effect due to its simple formulation and by updating the weights only on the last five timesteps. Table 3 also includes results from previous works using spiking models on the same datasets. For DVS Gesture, it can be seen that S-TLLR outperforms previous methods such as Xiao et al. (2022); Shrestha & Orchard (2018); Kaiser et al. (2020), in some cases with significantly less number of time-steps. In the case of DVS CIFAR10, S-TLLR demonstrates superior performance compared to the baseline with BPTT when the learning signal is utilized across all time steps ($T_l = 0$). Furthermore, S-TLLR surpasses the outcomes presented in Zheng et al. (2021); Fang et al. (2021b); Li et al. (2021), yet it lags behind others such as Xiao et al. (2022); Deng et al. (2022); Meng et al. (2022). However, studies such as Deng et al. (2022); Meng et al. (2022) showcase exceptional results primarily focused on static tasks without addressing temporal locality or memory efficiency during SNN training. Consequently, although serving as a reference, they do not fairly compare to S-TLLR. The most pertinent comparison lies with Xiao et al. (2022), which shares similar memory and time complexity with S-TLLR. Notably, S-TLLR ($T_l = 0$, $\alpha_{post} = -1$) exhibits a performance deficit of 0.67%. This difference is primarily attributed to the difference in batch size during training. While Xiao et al. (2022) uses a batch size of 128, we were constrained to 48 due to hardware limitations. To validate this point, we trained another model utilizing only causal terms: S-TLLR ($T_l = 0$, $\alpha_{post} = 0$), equating it to Xiao et al. (2022) considering the selection of STDP parameters and secondary activation function ($\Psi$) in DVS CIFAR10 experiments. The comparison reveals that S-TLLR ($\alpha_{post} = -1$) outperforms S-TLLR ($\alpha_{post} = 0$) (equivalent to Xiao et al. (2022)) under the same conditions, further corroborating the advantages of including non-causal terms ($\alpha_{post} \neq 0$) during training. Finally, when compared to BPTT, S-TLLR signifies a 5× memory reduction. By using the learning signal for the last five time steps, it effectively diminishes the number of multiply-accumulate (MAC) operations by 2.6× for DVS Gesture, and by 1.3× for DVS CIFAR10 and N-CALTECH101.

Table 3: Accuracy performance of methods on vision and audio datasets

| Method | Model | Time-steps (T) | Batch Size | Accuracy (mean±std) | # MAC[1] (×10⁹) | Memory[1] (MB) |
|---|---|---|---|---|---|---|
| DVS CIFAR10 | | | | | | |
| BPTT (Zheng et al., 2021) | ResNet-19 | 10 | - | 67.8% | - | - |
| BPTT (Fang et al., 2021b) | PLIF (7 layers) | 20 | 16 | 74.8% | - | - |
| TET (Deng et al., 2022) | VGG-11 | 10 | 128 | $83.17 \pm 0.15\%$ | - | - |
| DSR (Meng et al., 2022) | VGG-11 | 10 | 128 | $77.27 \pm 0.24\%$ | - | - |
| BPTT (Li et al., 2021) | ResNet-18 | 10 | - | $75.4 \pm 0.05\%$ | - | - |
| OTTT$_A$(Xiao et al., 2022) | VGG-9 | 10 | 128 | $76.27 \pm 0.05\%$ | - | - |
| **BPTT (baseline)** | VGG-9 | 10 | 48 | $75.44 \pm 0.76\%$ | 6.82 | 18.12 |
| **S-TLLR (Ours, $T_l = 5$, $\alpha_{post} = -1$ )** | VGG-9 | 10 | 48 | $73.93 \pm 0.62\%$ | 5.12 | 3.62 |
| **S-TLLR (Ours, $T_l = 0$, $\alpha_{post} = -1$ )** | VGG-9 | 10 | 48 | $75.6 \pm 0.10\%$ | 10.26 | 3.62 |
| **S-TLLR (Ours, $T_l = 0$, $\alpha_{post} = 0$ )** | VGG-9 | 10 | 48 | $74.8 \pm 0.15\%$ | 6.82 | 3.62 |
| **BPTT (baseline)** | ResNet18 | 10 | 48 | $72.68 \pm 0.87\%$ | 7.13 | 28.14 |
| **S-TLLR (Ours, $T_l = 5$)** | ResNet18 | 10 | 48 | $71.94 \pm 0.75\%$ | 5.12 | 5.62 |
| **S-TLLR (Ours, $T_l = 0$)** | ResNet18 | 10 | 48 | $74.5 \pm 0.64\%$ | 10.24 | 5.62 |
| DVS Gesture | | | | | | |
| SLAYER (Shrestha & Orchard, 2018) | SNN (8 layers) | 300 | - | $93.64 \pm 0.49\%$ | - | - |
| DECOLLE(Kaiser et al., 2020) | SNN (4 layers) | 1800 | 72 | $95.54 \pm 0.16\%$ | - | - |
| OTTT$_A$(Xiao et al., 2022) | VGG-9 | 20 | 16 | 96.88% | - | - |
| **BPTT (baseline)** | VGG-9 | 20 | 16 | $95.58 \pm 1.08\%$ | 6.06 | 16.13 |
| **S-TLLR (Ours)** | VGG-9 | 20 | 16 | $97.72 \pm 0.38\%$ | 2.27 | 1.61 |
| **BPTT (baseline)** | ResNet18 | 20 | 16 | $94.92 \pm 0.38\%$ | 6.34 | 25.03 |
| **S-TLLR (Ours)** | ResNet18 | 20 | 16 | $94.92 \pm 0.61\%$ | 2.27 | 2.50 |
| N-CALTECH101 | | | | | | |
| BPTT (She et al., 2022) | SNN (12 layers) | 10 | - | 71.2% | - | - |
| BPTT (Kim et al., 2023) | VGG-16 | 5 | 128 | 64.40% | - | - |
| **BPTT (baseline)** | VGG-9 | 10 | 16 | $65.92 \pm 0.82\%$ | 22.81 | 20.15 |
| **S-TLLR (Ours)** | VGG-9 | 10 | 16 | $66.058 \pm 0.92\%$ | 17.22 | 4.03 |
| **BPTT (baseline)** | ResNet18 | 10 | 16 | $60.89 \pm 0.89\%$ | 6.34 | 31.31 |
| **S-TLLR (Ours)** | ResNet18 | 10 | 16 | $61.65 \pm 0.99\%$ | 4.27 | 6.26 |
| SHD | | | | | | |
| ETLP (Quintana et al., 2023) | ALIF-RSNN | 100 | 128 | $74.59 \pm 0.44\%$ | - | - |
| OSTTP (Ortner et al., 2023) | LIF-RSNN | 100 | - | $77.33 \pm 0.8\%$ | - | - |
| BPTT (Bouanane et al., 2023) | LIF-RSNN | 100 | 128 | 83.41 | - | - |
| BPTT (Cramer et al., 2022) | LIF-RSNN | 100 | - | $83.2 \pm 1.3$ | - | - |
| **BPTT (baseline)** | LIF-RSNN | 100 | 128 | $70.57 \pm 0.96$ | 0.054 | 0.961 |
| **S-TLLR$_{BP}$ (Ours)** | LIF-RSNN | 100 | 128 | $78.24 \pm 1.84\%$ | 0.096 | 0.019 |
| **S-TLLR$_{DFA}$ (Ours)** | LIF-RSNN | 100 | 128 | $74.60 \pm 0.52\%$ | 0.096 | 0.019 |

[1]: # MAC and Memory are estimated for a batch size of 1 following the equations (23) and (22) described in the Section 4.3.

[2]: Previous studies' accuracy values are provided as reported in their respective original papers.

### 5.2.2 Audio Classification

In order to set a baseline, we train the same RSNN with the same hyperparameters using BPTT for five trials, and with the following STPD parameters $(0.5, 1, 1, 1)$. As shown in Table 3, the model trained with S-TLLR outperforms the baseline trained with BPTT. The result shows the capability of S-TLLR to achieve high performance and generalization. One reason why the baseline does not perform well, as suggested in Cramer et al. (2022), is that RSNN trained with BPTT quickly overfits. This also highlights a nice property of S-TLLR. Since it has a simpler formulation than BPTT, it can avoid overfitting, resulting in a better generalization. However, note that works such as Cramer et al. (2022); Bouanane et al. (2023) can achieve better performance after carefully selecting the hyperparameters and using data augmentation techniques. In comparison with such works, our method still shows competitive performance with the advantage of having a $50\times$ reduction in memory.

Furthermore, we compared our results with Quintana et al. (2023); Ortner et al. (2023), which uses the same RSNN network structure with LIF and ALIF (LIF with adaptive threshold) neurons and temporal local learning rules. Table 3 shows that using S-TLLR with BP for the learning signal results in better

Table 4: Comparison of the Average End-Point Error (AEE) on the MVSEC (Zhu et al., 2018b) dataset [AEE lower is better]

| Models | Training Method | Type | OD1 AEE | IF1 AEE | IF2 AEE | IF3 AEE | AEE Sum |
|---|---|---|---|---|---|---|---|
| **FSFN**$_{\alpha_{post} = -0.2}$ **(Ours)** | S-TLLR | Spiking | 0.50 | 0.76 | 1.19 | 1.00 | 3.45 |
| **FSFN**$_{\alpha_{post} = 0.2}$ **(Ours)** | S-TLLR | Spiking | 0.54 | 0.78 | 1.28 | 1.09 | 3.69 |
| **FSFN**$_{\alpha_{post} = 0}$ **(Ours)** | S-TLLR | Spiking | 0.50 | 0.77 | 1.25 | 1.08 | 3.60 |
| **FSFN (baseline)** | BPTT | Spiking | 0.45 | 0.76 | 1.17 | 1.02 | 3.40 |
| Apolinario et al. (2023) | BPTT | Spiking | 0.51 | 0.82 | 1.21 | 1.07 | 3.61 |
| Kosta & Roy (2023) | BPTT | Spiking | 0.44 | 0.79 | 1.37 | 1.11 | 3.78 |
| Hagenaars et al. (2021) | BPTT | Spiking | 0.45 | 0.73 | 1.45 | 1.17 | 3.80 |
| Zero prediction | - | - | 1.08 | 1.29 | 2.13 | 1.88 | 6.38 |

performance than those obtained with other temporal local learning rules, with the advantage of having a linear memory complexity instead of squared. Moreover, using DFA to generate the learning signal results in competitive performance with the advantage of being local in both time and space.

### 5.2.3 Event-based Optical Flow

The optical flow estimation is evaluated using the average endpoint error (AEE) metric that measures the Euclidean distance between the predicted flow ($\mathbf{y}^{\text{pred}}$) and ground truth flow ($\mathbf{y}^{\text{gt}}$) per pixel. For consistency, this metric is computed only for pixels containing events ($P$), similar to Apolinario et al. (2023); Kosta & Roy (2023); Lee et al. (2020); Zhu et al. (2018a; 2019), given by the following expression:

$$\text{AEE} = \frac{1}{P} \sum_P \|\mathbf{y}^{\text{pred}}_{i,j} - \mathbf{y}^{\text{gt}}_{i,j}\|_2 \tag{24}$$

For this experiment, we trained a Fully-Spiking FlowNet (FSFN) model, discussed in Appedix A, with S-TLLR using the following STDP parameters $(\lambda_{post}, \lambda_{pre}, \alpha_{pre}) = (0.5, 0.8, 1)$ and $\alpha_{post} = [-0.2, 0.2, 0]$. The models were trained during 100 epochs using the Adam optimizer with a learning rate of 0.0002, a batch size of 8, and with the learning signal obtained from the photometric loss just for the last time step ($T_l = 9$). As it is shown in Table 4, the FSFN model trained using S-TLLR with $\alpha_{post} = -0.2$ shows a performance close to the baseline implementation trained with BPTT. Although we mainly compared our model with BPTT, to take things into perspective, we include results from other previous works. Among the spiking models, our model trained with S-TLLR has the second-best average performance (AEE sum) in comparison with such spiking models of similar architecture and size trained with BPTT (Apolinario et al., 2023; Kosta & Roy, 2023; Hagenaars et al., 2021). The results indicate that our method achieves high performance on a complex spatio-temporal task, such as optical flow estimation, with 5× less memory and a 6.6× reduction in the number of MAC operations by just updating the model in the last time step.

## 6 Conclusion

Our proposed learning rule, S-TLLR, can achieve competitive performance in comparison to BPTT on several event-based datasets with the advantage of having a constant memory requirement. In contrast to BPTT (or other temporal learning rules) with higher memory requirements $O(Tn)$ (or $O(n^2)$), S-TLLR memory is just proportional to the number of neurons $O(n)$. Moreover, in contrast with previous works that are derived from BPTT as approximations, and therefore using only causal relations in the spike timing, S-TLLR explores a different direction by leveraging causal and non-causal relations based on a generalized parametric STDP equation. We have experimentally demonstrated on several event-based datasets that including such non-causal relations can improve the SNN performance in comparison with temporal local learning rules using just causal relations. Also, we could observe that tasks where spatial information is predominant, such as DVS CIFAR-10, DVS Gesture, N-CALTECH101, and MVSEC, benefit from causality ($\alpha_{post} = -1$). In contrast, tasks like SHD, where temporal information is predominant, benefit from synchrony ($\alpha_{post} = 1$).

Moreover, by computing the learning signal just for the last few time steps, S-TLLR reduces the number of MAC operations in the range of $1.3\times$ to $6.6\times$. In summary, S-TLLR can achieve high performance while being memory-efficient and requiring only information locally in time, therefore enabling online updates.

**Acknowledgments**

This work was supported in part by the Center for CoDesign of Cognitive Systems (CoCoSys), one of seven centers in JUMP 2.0, funded by the Semiconductor Research Corporation (SRC) and DARPA, in part by the MicroE4AI program of IARPA, the DoE Microelectronics program, the National Science Foundation, and Intel Corporation.

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

## A  Datasets and experimental setup

We conducted experiments on various event-based datasets, including DVS Gesture (Amir et al., 2017), N-CALTECH101 (Orchard et al., 2015), DVS CIFAR-10 (Li et al., 2017), SHD (Cramer et al., 2022), and MVSEC (Zhu et al., 2018b). These datasets encompass a wide range of applications, such as image and gesture recognition, audio classification, and optical flow estimation. In this section, we outline the experimental setup employed in Section 5, covering SNN architectures, dataset preprocessing, and loss functions.

### A.1  Network architectures

For experiments on image and gesture recognition, we use a VGG-9 model with the following structure: 64C3-128C3-AP2S2-256C3-256C3-AP2S2-512C3-512C3-AP2S2-512C3-512C3-AP2S2-FC. In this notation, '64C3' signifies a convolutional layer with 64 output channels and a 3x3 kernel, 'AP2S2' represents average-pooling layers with a 2x2 kernel and a stride of 2, and 'FC' denotes a fully connected layer. In addition, instead of

batch normalization, we use weight standardization Qiao et al. (2019) similar to Xiao et al. (2022). Also, the leak factor and threshold for the LIF models (1) are $\gamma = 0.5$ and $v_{\mathrm{th}} = 0.8$, respectively.

For experiments with the SHD dataset, we employ a recurrent SNN (RSNN) consisting of one recurrent layer with 450 neurons and a leaky integrator readout layer with 20 neurons, following a similar configuration as in Ortner et al. (2023); Quintana et al. (2023). Both layers are configured with a leak factor of $\gamma = 0.99$, and the recurrent LIF layer's threshold voltage is set to $v_{\mathrm{th}} = 0.8$.

We utilize the Fully-Spiking FlowNet (FSFN) for optical flow estimation, as introduced by Apolinario et al. (2023). The FSFN adopts a U-Net-like architecture characterized by performing binary spike computation in all layers. Notably, we enhance the model by incorporating weight standardization (Qiao et al., 2019) in all convolutional layers. Additionally, our training approach involves ten time steps without temporal input encoding, in contrast to Apolinario et al. (2023) that uses the encoding method proposed by Lee et al. (2020). Also, the leak factor and threshold of the LIF neurons used for our FSFN are 0.88 and 0.6, respectively.

## A.2 Data pre-processing

Here we describe the data pre-processing for each dataset:

- DVS Gesture: the dataset contains 11 hand gesture categories from 29 subjects under 3 illumination conditions recorded with a Dynamic Vision Sensor (DVS) with a resolution of 128×128 pixels. The recordings were split into sequences of 1.5 seconds of duration, and the events were accumulated into 20 bins (event frames), with each bin having a 75 ms time window. Then, the event frames were resized to a size of $32 \times 32$, while maintaining the positive and negative polarities as channels.

- N-CALTECH: is a spiking version of the original frame-based CALTECH101 dataset. It was produced by recording the static images of the CALTECH101 dataset displayed on a LCD monitor using an ATIS sensor (an event-based camera) while moving it. The recordings were wrapped into ten bins with the same time window size for each bin (30 ms). Then, the event frames were resized to a dimension of $60 \times 45$.

- DVS CIFAR10: is a spiking version of 10000 samples from the frame-based CIFAR10 dataset recorded with a event-based camera at a resolution of 128×128 pixels. The recordings were wrapped into ten bins with the same time window size for each bin. Then, the event frames were resized to a dimension of $48 \times 48$. Additionally, a random crop with a padding of 4 was used for data augmentation.

- SHD: is an audio-classification dataset consisting of spoken digits ranging from 0 to 9 in both English and German. The audio waveforms were converted into spike trains using an artificial model of the inner ear auditory system. The events (spikes) in each sequence were wrapped into 100 bins, each with a time window duration of 10 ms. No data augmentation techniques were used for SHD.

- MVSEC: is a event-based dataset used for training and evaluating optical flow predictions. It contains stereo event-based data collected in various environments, including indoor flying and outdoor driving, along with the corresponding ground truth optical flow. The events between two consecutive grayscale frames were wrapped into ten bins, keeping the negative and positive polarities as channels. Then, the event frames are fed to the SNN model sequentially, similar to the approach in Apolinario et al. (2023).

## A.3 Loss functions and secondary activation functions ($\Psi$)

For image, gesture, and audio classification tasks, we utilized cross-entropy (CE) loss and computed the learning signal using (12) with ground truth labels ($\mathbf{y}^*$). In contrast, for optical flow, we employed a self-supervised loss based on photometric and smooth loss, as detailed in Equation (5) in Lee et al. (2020).

Regarding the generation of the learning signal ($\boldsymbol{\delta}$), in the context of image and gesture recognition, it is exclusively generated for the final five time steps ($T_l = 5$). For audio classification, we employ $T_l = 90$, while

for optical flow, we use $T_l = 1$. This setting reduces the number of computations compared to BPTT by factors of $4\times$, $1.1\times$, and $10\times$, respectively.

Finally, we consider the following secondary activation functions for the computation of the eligibility traces (11):

$$\Psi(u_i[t]) = \frac{1}{(100|u_i[t] - v_{\text{th}}| + 1)^2} \tag{25}$$

$$\Psi(u_i[t]) = 0.3 \times \max(1.0 - |u_i[t] - v_{\text{th}}|, 0) \tag{26}$$

$$\Psi(u_i[t]) = 4 \times \text{sigmoid}(u_i[t] - v_{\text{th}})(1 - \text{sigmoid}(u_i[t] - v_{\text{th}}) \tag{27}$$

$$\Psi(u_i[t]) = \frac{1}{1 + (10(u_i[t] - v_{\text{th}}))^2} \tag{28}$$

Here, $\max(a, b)$ returns the maximum between $a$ and $b$, and $|\cdot|$ represents the absolute value function. These activation functions (25), (26), (27), and (28) are specifically used for SHD, DVS Gesture, DVS CIFAR10, and MVSEC, respectively.

Note that the secondary activation function plays a role similar to that of the surrogate gradient in BPTT, so the selected secondary functions were adapted from previous works that utilized similar functions in BPTT schemes. Specifically, (25) was adapted from Ortner et al. (2023), (26) from Apolinario et al. (2023), (27) from Wu et al. (2018), and (28) from Hagenaars et al. (2021).

### A.4  Experiments Setup

### A.4.1  Experimental setup: effects of non-causal terms on learning

For the experiments conducted on DVS Gesture, N-CALTECH101, and DVS CIFAR10 datasets, we trained a VGG9 model for 30, 30, and 300 epochs, respectively. In all these experiments, we utilized the Adam optimizer with a learning rate of 0.001. Additionally, the learning signal was presented only during the last five time steps, meaning the model was updated and the error computed five times per sample. For these experiments, the parameters $\lambda_{post}$, $\lambda_{pre}$, and $\alpha_{pre}$ were kept constant, while $\alpha_{post}$ was varied to simulate different scenarios: the absence of non-causal terms ($\alpha_{post} = 0$), the inclusion of non-causal terms ($\alpha_{post} = +1$), and the subtraction of non-causal terms ($\alpha_{post} = -1$). This approach allowed us to evaluate the impact of non-causal terms on the learning rule.

For the experiments on the SHD dataset, we trained a recurrent spiking neural network (SNN) model for 200 epochs. In this setup, we used the Adam optimizer with a learning rate of 0.0002 and a batch size of 128. Similar to the previous experiments, $\alpha_{post}$ was varied among three values ($-1$, 0, and $+1$) to assess the effect of non-causal terms while keeping all other parameters constant.

## B  Computational analysis of BPTT and S-TLLR

In Section 4.3, we discussed the computational improvements of S-TLLR over BPTT. Here, we further detailed the procedure for obtaining (34), and present an example of real GPU memory usage with BPTT and S-TLLR.

### B.1  BPTT analysis

To analyze BPTT, we follow a similar analysis as Bellec et al. (2020). Here, we will utilize a three-layer feedforward SNN as illustrated in Fig. B. Our analysis is based on a regression problem with the target denoted as $\boldsymbol{y}^*$ across $T$ time steps, and our objective is to compute the gradients for the weights of the first layer ($\boldsymbol{w}^{(1)}$). The Mean Squared Error (MSE) loss function ($\mathcal{L}$) is defined as follows:

$$\mathcal{L} = \frac{1}{2}\|\boldsymbol{y}^* - \sum_{t=0}^{T} \boldsymbol{y}^{(3)}[t]\|_2^2 \tag{29}$$

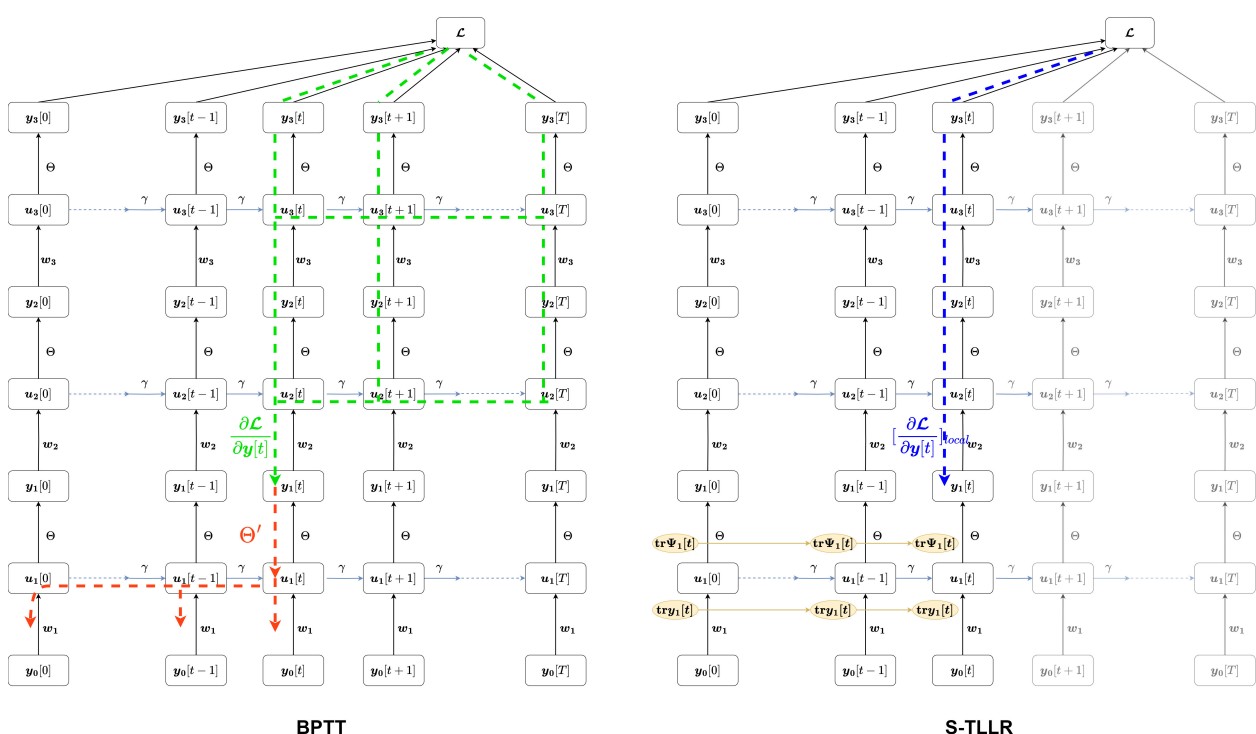

Figure 2: Comparison of weight update computation of three feed-forward spiking layers for BPTT and S-TLLR. The spiking layers are unrolled over time for both algorithms while showing the signals involved in the weight updates. The top-down learning signal for BPTT ($\frac{\partial \mathcal{L}}{\partial y[t]}$) is shown in green, note that at time step $t$ this learning signal depends on future time steps. In contrast, for S-TLLR the learning signal ($[\frac{\partial \mathcal{L}}{\partial y[t]}]_{local}$) and all the variables involved are computed with information available only the time step $t$.

Here $\boldsymbol{y}^{(3)}[t]$ is the output of the third layer (last layer) at the time step ($t$). Then, the gradients with respect to $\boldsymbol{w}^{(1)}$ can be computed as follows:

$$\frac{d\mathcal{L}}{d\boldsymbol{w}^{(1)}} = \sum_{t=0}^{T} \frac{\partial \mathcal{L}}{\partial \boldsymbol{u}^{(1)}[t]} \frac{\partial \boldsymbol{u}^{(1)}[t]}{\partial \boldsymbol{w}^{(1)}} \tag{30}$$

If we expand the term $\frac{\partial \mathcal{L}}{\partial \boldsymbol{u}^{(1)}[t]}$, we obtain the following expression:

$$\frac{\partial \mathcal{L}}{\partial \boldsymbol{u}^{(1)}[t]} = \frac{\partial \mathcal{L}}{\partial \boldsymbol{y}^{(1)}[t]} \frac{\partial \boldsymbol{y}^{(1)}[t]}{\partial \boldsymbol{u}^{(1)}[t]} + \frac{\partial \mathcal{L}}{\partial \boldsymbol{u}^{(1)}[t+1]} \frac{\partial \boldsymbol{u}^{(1)}[t+1]}{\partial \boldsymbol{u}^{(1)}[t]} \tag{31}$$

Note that the right-hand side contains the term $\frac{\partial \mathcal{L}}{\partial \boldsymbol{u}^{(1)}[t+1]}$ that could be further expanded. If we apply (31) recursively, replace it on (30), and factorize the $\frac{\partial \mathcal{L}}{\partial \boldsymbol{y}^{(1)}[t]}$ terms, we obtain the following expression:

$$\frac{d\mathcal{L}}{d\boldsymbol{w}^{(1)}} = \sum_{t=0}^{T} \sum_{t'=t}^{T} \frac{\partial \mathcal{L}}{\partial \boldsymbol{y}^{(1)}[t']} \frac{\partial \boldsymbol{y}^{(1)}[t']}{\partial \boldsymbol{u}^{(1)}[t']} \left( \prod_{k=0}^{t'-t} \frac{\partial \boldsymbol{u}^{(1)}[t'-k]}{\partial \boldsymbol{u}^{(1)}[t'-k-1]} \right) \frac{\partial \boldsymbol{u}^{(1)}[t]}{\partial \boldsymbol{w}^{(1)}} \tag{32}$$

By doing a change of variables between $t$ and $t'$, we can rewrite (32) as:

$$\frac{d\mathcal{L}}{d\boldsymbol{w}^{(1)}} = \sum_{t'=0}^{T} \frac{\partial\mathcal{L}}{\partial\boldsymbol{y}^{(1)}[t']} \frac{\partial\boldsymbol{y}^{(1)}[t']}{\partial\boldsymbol{u}^{(1)}[t']} \sum_{t=0}^{t'} \Big(\prod_{k=0}^{t'-t} \frac{\partial\boldsymbol{u}^{(1)}[t'-k]}{\partial\boldsymbol{u}^{(1)}[t'-k-1]}\Big) \frac{\partial\boldsymbol{u}^{(1)}[t]}{\partial\boldsymbol{w}^{(1)}} \tag{33}$$

Further replacing the LIF equation (1) on (33), we obtain:

$$\frac{d\mathcal{L}}{d\boldsymbol{w}^{(1)}} = \sum_{t'=0}^{T} \frac{\partial\mathcal{L}}{\partial\boldsymbol{y}^{(1)}[t']} \Theta'(\boldsymbol{u}^{(1)}[t']) \sum_{t=0}^{t'} \gamma^{t'-t}\boldsymbol{y}^{(0)}[t] \tag{34}$$

$$\Delta\boldsymbol{w}^{(1)}[t'] = \frac{\partial\mathcal{L}}{\partial\boldsymbol{y}^{(1)}[t']} \Theta'(\boldsymbol{u}^{(1)}[t']) \sum_{t=0}^{t'} \gamma^{t'-t}\boldsymbol{y}^{(0)}[t] \tag{35}$$

The total update of the weight for the first layer is shown in (34) where at each time step, the synaptic updated is represented by (35). Here, it can be seen that the contribution of any time step ($t'$) has two components, $\Theta'(\boldsymbol{u}^{(1)}[t']) \sum_{t=0}^{t'} \gamma^{t'-t}\boldsymbol{y}^{(0)}[t]$ that at time-step $t'$ depends only on previous information $(0, 1, ..., t'-1, t')$, and an learning signal ($\frac{\partial\mathcal{L}}{\partial\boldsymbol{y}^{(1)}[t']}$) depends on information of future time steps $(t'+1, t'+2, ..., T)$. Those components are visualized in Fig. B. Since the error signal depends on future time steps it can not be computed locally in time, that is with information available only at the time step ($t'$). Therefore, BPTT is not a temporal local learning rule.

## B.2 Example of real GPU memory usage

This section shows the substantial memory demands associated with BPTT and their correlation with the number of time steps ($T$). To illustrate this point, we employ a simple regression problem utilizing a synthetic dataset ($x, y^*$), where $x$ denotes a vector of dimension 1000 and $y^*$ represents a scalar value. The batch size used is 512. The structure of the SNN model comprises five layers, structured as follows: 1000FC-1000FC-1000FC-1000FC-1FC. In addition, the loss is computed solely for the final time step as $\mathcal{L} = (y[T]-y^*)^2$. This evaluation is conducted on sequences of varying lengths (10, 25, 50, 100, 200, and 300). To simplify, these sequences are generated by repeating the same input ($x$) multiple times. Throughout these experiments, we utilized an NVIDIA GeForce GTX 1060 and recorded the peak memory allocation. The obtained results are visualized in Fig. B.2. These results distinctly highlight how the memory usage of BPTT scales linearly with the number of time steps ($T$), while S-TLLR remains constant.

# C Additional experiments on the effects of STDP parameters on learning

## C.1 Effects of causality and non-causality factors using DFA

In Section 5.1 of the main text, we examined the impact of introducing non-causal terms in the computation of the instantaneous eligibility trace (10) when using error-backpropagation (BP) to generate the learning signal. In this section, we conduct a similar experiment, but this time, we employ direct feedback alignment (DFA) for the learning signal generation.

As in Section 5.1, we vary the values of $\alpha_{post}$, including $-1$, $0$, and $1$, to assess the effect of non-causal terms. Interestingly, when the learning signal is produced via random feedback with DFA, there is no significant difference observed when including non-causal terms or not. For example, in the RSNN model, using $\alpha_{post} = 1$ yields slightly better performance, as depicted in Fig. C.1, but the difference is marginal. Similarly, for the VGG9 model, the performance of $\alpha_{post} = 1$ and $\alpha_{post} = 0$ is comparable and superior to $\alpha_{post} = -1$, as shown in Fig. C.1. This suggests that a more precise learning signal, such as BP, may be necessary to fully exploit the benefits of non-causal terms.

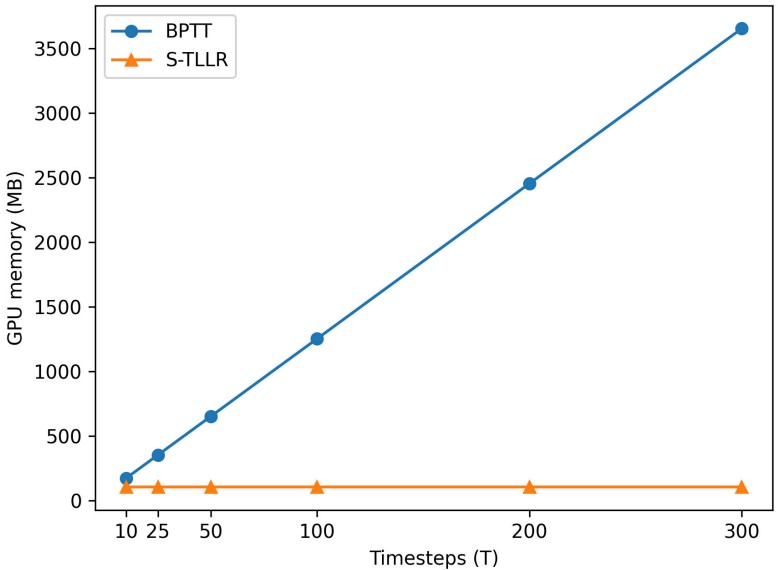

Figure 3: GPU memory usage for BPTT and S-TLLR for a five-layer fully connected SNN models with a different number of time steps ($T$).

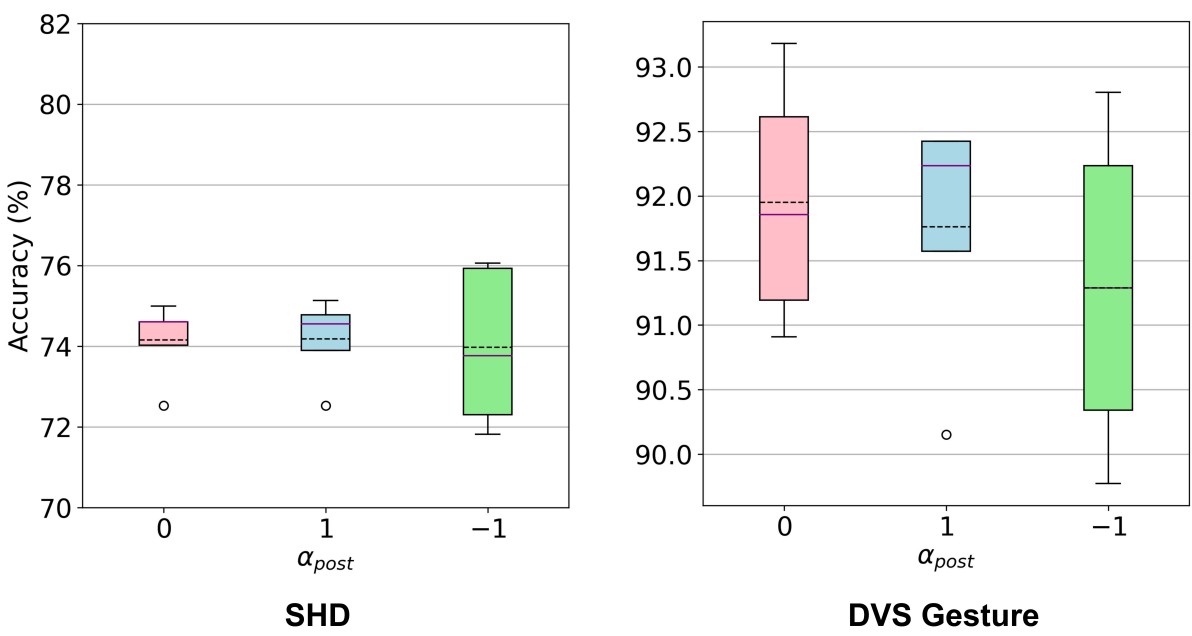

Figure 4: Evaluating the effect of $\alpha_{\text{post}}$ on DVS Gesture and SHD datasets using DFA for learning signal generation. Constant STDP parameters for DVS Gesture are $(\lambda_{post}, \lambda_{pre}, \alpha_{pre}) = (0.2, 0.75, 1)$, and for SHD, they are $(\lambda_{post}, \lambda_{pre}, \alpha_{pre}) = (0.5, 1, 1)$. The solid purple line represents the median value, and the dashed black line represents the mean value averaged over five trials.

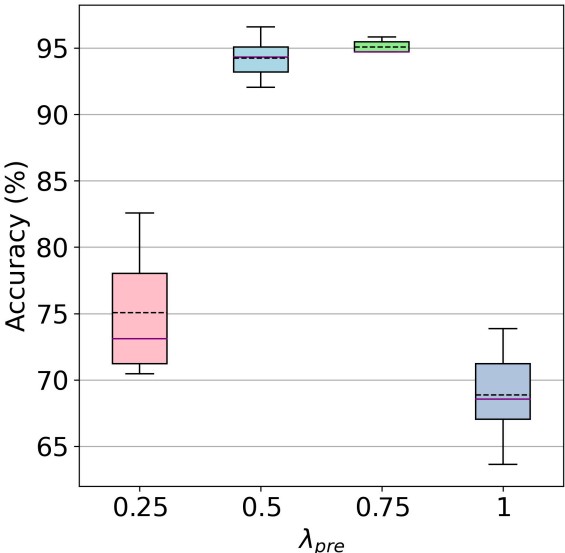

Figure 5: Effects of using a decaying factor, $\lambda_{\text{pre}}$, different from the leak spiking parameter ($\gamma = 0.5$) to compute the causal term on the eligibility trace, with constant parameters $(\lambda_{post}, \alpha_{post}, \alpha_{pre}) = (0.2, -1, 1)$ when the learning signal is generated using BP. Plots are based on five trials. The solid purple line represents the median value, and the dashed black line represents the mean value.

## C.2 Effects of $\lambda_{\text{pre}}$ on the learning

In this section, we explore the impact of using a $\lambda_{\text{pre}}$ value different from the leak parameter $\gamma$ of the LIF model (1) on the S-TLLR eligibility trace (10). This parameter, which controls the decay factor of the input trace, offers an opportunity to optimize the model's performance. While previous works aiming to approximate BPTT Xiao et al. (2022); Bellec et al. (2020); Bohnstingl et al. (2022) set $\lambda_{\text{pre}}$ equal to the leak factor ($\gamma$), our experiments, as depicted in Fig. C.1, suggest that using a slightly higher $\lambda_{\text{pre}}$ can lead to improved average accuracy performance.

## C.3 Ablation studies on the secondary activation function ($\Psi$)

In this section, we present ablation studies on the secondary activation function using the DVS Gesture, NCALTECH101, and SHD datasets, with the same configurations used for experiments on Table 2. The results are presented in Table 5, where it can be seen that independent of $\Psi$ function using a non-zero $\alpha_{post}$ result in better performance than using $\alpha_{post} = 0$.

Table 5: Ablation studies on the secondary activation function $\Psi$ [Accuracy (mean±std) reported over 5 trials ]

| $\Psi$ function | Model | T | $T_l$ | $(\lambda_{post}, \lambda_{pre}, \alpha_{pre})$ | $\alpha_{post} = 0$ | $\alpha_{post} = +1$ | $\alpha_{post} = -1$ |
|---|---|---|---|---|---|---|---|
| DVS128 Gesture | | | | | | | |
| (25) | VGG9 | 20 | 15 | (0.2, 0.75, 1) | $73.61 \pm 2.92\%$ | $\mathbf{81.89 \pm 5.13}\%$ | $65.45 \pm 2.92\%$ |
| (26) | VGG9 | 20 | 15 | (0.2, 0.75, 1) | $94.61 \pm 0.73\%$ | $94.01 \pm 1.10\%$ | $\mathbf{95.07 \pm 0.48}\%$ |
| (27) | VGG9 | 20 | 15 | (0.2, 0.75, 1) | $94.46 \pm 0.45\%$ | $94.46 \pm 0.45\%$ | $\mathbf{95.85 \pm 0.66}\%$ |
| NCALTECH101 | | | | | | | |
| (25) | VGG9 | 10 | 5 | (0.2, 0.5, 1) | $35.11 \pm 0.40\%$ | $\mathbf{37.19 \pm 1.17}\%$ | $28.53 \pm 0.56\%$ |
| (26) | VGG9 | 10 | 5 | (0.2, 0.5, 1) | $63.34 \pm 0.96\%$ | $54.07 \pm 2.37\%$ | $\mathbf{66.05 \pm 0.92}\%$ |
| (27) | VGG9 | 10 | 5 | (0.2, 0.5, 1) | $62.24 \pm 1.22\%$ | $53.42 \pm 1.50\%$ | $\mathbf{66.33 \pm 0.86}\%$ |
| SHD | | | | | | | |
| (25) | RSNN | 100 | 10 | (0.5, 1, 1) | $77.09 \pm 0.33\%$ | $\mathbf{78.23 \pm 1.84}\%$ | $74.69 \pm 0.47\%$ |
| (26) | RSNN | 100 | 10 | (0.5, 1, 1) | $76.25 \pm 0.44\%$ | $\mathbf{76.28 \pm 0.25}\%$ | $74.40 \pm 0.59\%$ |
| (27) | RSNN | 100 | 10 | (0.5, 1, 1) | $75.22 \pm 0.79\%$ | $\mathbf{76.29 \pm 0.25}\%$ | $74.46 \pm 0.65\%$ |

