# OpenReview forum: "S-TLLR: STDP-inspired Temporal Local Learning Rule for Spiking Neural Networks"
_TMLR — Accepted by TMLR_

### Review · Reviewer_FrTj · 2024-05-16

**Summary Of Contributions:**

This paper presents S-TLLR, a three-factor temporal local learning rule for spiking neural network training on even-based tasks. Besides the causal relations between the timing of pre- and post-synaptic activities, which is commonly used by existing training methods, the authors extensively take the non-causal relations into account to further enhance the training performance. The evaluation is conducted on various datasets and different tasks, and the experiment results show that the proposed learning method obtains competitive performance compared to BPTT and other learning rules.

**Audience:**

Yes

**Claims And Evidence:**

Yes

**Requested Changes:**

The idea of this manuscript is interesting; however, some technical details and experiment results lack clarifications and justifications.  Please see the comments below:

- My major concern is the effects of the non-causal factor. From Table 2, only minor improvements are observed on 4 datasets, and the value of the factor has different impacts on different tasks. To me, the results are not convincing that the non-causal can indeed benefit the S-TLLR.

- The authors may add some brief introductions to the datasets used for in experiments. There is a lack of detailed explanation of the experiment setup, which makes the results somehow hard to interpret.

- The performance of S-TLLR does not outperform BPTT and other methods (Table 3). The author may elaborate more on the advantages and motivations of adapting the proposed method instead of the existing ones.

- The presentation requires further improvement; I noticed several typos and grammar issues (e.g., Section 5.1, non-casual -> non-causal).

**Strengths And Weaknesses:**

Strengths:

+ Large-scale experiments across different datasets and models.
+ The proposed method has reduced time and memory complexity.


Weaknesses:

- The effectiveness of non-casual terms requires further justification.
- The performance of S-TLLR does not outperform other methods on 4 datasets.

---

> ### Author Response · Authors · 2024-08-22
>
> We would like to thank the reviewer for reading our work and providing feedback that will enhance our manuscript. We would now like to address the points highlighted by the reviewer and outline the actions we take.
> - **Effects of Non-Causal Factors:** We appreciate your concern regarding the non-causal factors. We would like to emphasize that the improvements in accuracy are significant, ranging from 0.4% to 4% across the four tasks when compared to S-TLLR using only causal terms ($\alpha_{post}=0$). It is natural that the accuracy is primarily driven by the causal term in the spiking activity. However, as shown in Table 2, incorporating non-causal information ($\alpha_{post}=+1$ or $\alpha_{post}=-1$) enhances model performance. The different impacts of the factor on various tasks are expected because $\alpha_{post}=+1$ and $\alpha_{post}=-1$ represent different forms of STDP. Specifically, $\alpha_{post}=-1$ models an antisymmetric STDP form, rewarding causal and blaming non-causal terms, whereas $\alpha_{post}=+1$ models a symmetric STDP, rewarding or blaming both terms equally in favor of synchrony. This approach was inspired by recent findings from Anisimova et al. (2022), which suggest that STDP initially favors causality but evolves to reward both causal and non-causal relations over time. Our results in Table 2 indicate that for tasks with significant spatial information, an antisymmetric STDP function ($\alpha_{post}=-1$) offers better performance, while symmetric STDP ($\alpha_{post}=+1$) benefits tasks where temporal information is predominant. This observation is further supported by the experiments with Optical Flow (a task with significant spatial information) in Table 4, where the antisymmetric STDP function ($\alpha_{post}=-1$) performs better. Overall, including non-causal terms benefits the performance of the SNN model.
> - **Experimental Setup:** Thank you for this suggestion. We updated the manuscript to expand the presentation and details of the experimental setup provided in Appendix A.
> - **Performance Comparison:** We would like to clarify that, as reported in Table 3, S-TLLR outperforms BPTT (baseline) in all four tasks. We set a baseline with BPTT to ensure a fair comparison by using the same parameters for both S-TLLR and BPTT. To provide context, we also report the performance of previous works. However, it is important to note that implementation details and specific features of each method can make fair comparison challenging. Under the same conditions, S-TLLR can outperform BPTT while offering lower and constant memory requirements, as shown in Table 3. The advantages of using S-TLLR over BPTT include comparable performance at a significantly lower memory cost, which is crucial for deploying SNNs on low-power edge systems with on-device learning capabilities. Many works using BPTT propose innovations (e.g., spiking models, architectures, loss functions) that can complement S-TLLR. Other works are mainly designed for static tasks and do not cover the same spectrum of applications.
> - **Presentation Improvements:** Thank you for pointing this out. We updated the manuscript to correct typos and grammar issues.

---

### Review · Reviewer_CiSd · 2024-05-27

**Summary Of Contributions:**

This work presents a new local learning rule for spiking neural networks. By design, it has low memory and compute complexities. Further, unlike the standard approach (backprop-through-time), the proposed method includes "non-causal relations" in its computations. Its effectiveness on event-based datasets is experimentally shown.

**Audience:**

Yes

**Broader Impact Concerns:**

-

**Claims And Evidence:**

Yes

**Requested Changes:**

- $x_j[t]$ in (1) is not defined near (1).
- The comment "$\gamma$ is the leak factor that reduces the membrane potential over time" should be elaborated as it's not clear immediately.
- Based on section 4.2.1, $u_i[t]$ in (10) is not computed recursively. Then, how is it computed?
- A discussion on $\Psi$ would be great.
- About the note "Note that the backpropagation occurs through the layers and not in time"; what equation implies that? Is it due to (12) or breaking the recurrence?
- Highlighting the results in bold would be nice.
- Ideally, one can compute the memory footprints and the number of floating point operations of all methods (e.g., for one epoch) and add these numbers to a separate column in Table 3. This would let us compare the presented approach and all other baselines for which #MAC and Memory is not reported.

**Strengths And Weaknesses:**

### Strengths
- The biggest strength of the presented approach is the significant improvements in computational complexity and memory use.
- I liked the writing and presentation. No major change is needed but please see my comments in the "requested changes" section.
- I did not find any mistake in the computations/algorithm.

### Weaknesses
- The biggest weakness is that the approach seems a little hand-wavy... More specifically, as far as I understand, it is a combination of BPTT and STDP. While these two are based on clear learning principles, which are standard gradient descent optimization and inspirations from neural learning systems, the presented approach does not come from the first principles. For example,
  - The impact of setting $\beta=0$ (from a learning perspective) is not discussed. Is this a simple trick to handle memory inefficiency? Wouldn't breaking the recurrence completely alter the learning signal?
  - Likewise, what is the role of $\Psi$? Is it a non-linearity to make the whole framework more expressive or does it play a more crucial role? What would it be its counterpart in BPTT or STDP or alternative approaches?
- The results don't seem very impressive and are also slightly inconsistent. Is the presented approach now more advantageous in general? What would be a simple principle (e.g., based on the amount and modality of data, difficulty of problem, etc) that would help us choose this approach over others?

---

> ### Author Response · Authors · 2024-08-22
>
> We would like to thank reviewer for the constructive feedback. We appreciate the opportunity to address the points raised and outline the revisions and clarifications we made:
> - **Regarding the first point in the weakness section:** We want to clarify that our proposed learning rule is not a combination of BPTT and STDP. Instead, it is a three-factor learning rule where the eligibility traces are inspired by the STDP mechanism. Evidence suggests that three-factor learning rules with eligibility traces occur in biological neural plasticity [1]. Thus, our work draws inspiration from these biological mechanisms (both three-factor learning and STDP) to design a learning method for artificial spiking neural networks (SNNs).
>     - **Regarding the use of $\beta=0$:** We would like to emphasize that eligibility traces are not learning signals on their own but rather signals that track the importance of particular synaptic connections over time based on pre- and post-synaptic activity. Our experiments indicate that maintaining traces of pre- and post-synaptic activity captures this temporal information effectively.
>     - **Regarding the role of $\Psi$:** The function $\Psi$ indeed makes the learning rule more expressive. Modifying the parameters of this function can enhance the model’s efficiency. In BPTT, the analog would be a surrogate gradient function (an approximation of the gradient of the firing function in SNNs), while in STDP, it would be the firing function of the SNNs.
> -	**On performance and comparison:** As reported in Table 3, S-TLLR performs comparably to or even outperforms BPTT (the baseline) in all four tasks. We used BPTT as a baseline to ensure a fair comparison by employing the same parameters for both S-TLLR and BPTT. In terms of both memory efficiency and performance, S-TLLR offers advantages over BPTT. The key factor in choosing our proposed learning rule over others is the amount of memory available in the deployment system. We designed S-TLLR with the intent of use in on-device learning scenarios on edge devices, where memory is a critical constraint. Therefore, our memory-efficient learning rule is more suitable than BPTT for such scenarios.
> -	**Rearrangement of figures and tables:** We rearranged the figures and tables so that the usage and definitions of variables are more closely aligned.
> -	**Computation of $u_i[t]$:** There seems to be a misinterpretation here. The variable $u_i[t]$ represents the membrane potential of the SNNs and is always computed recursively due to the leak factor ($\gamma$), as defined in equation (1). Section 4.2.1 mainly discusses recurrent synaptic connections ($w^{rec}$), which are not present in equation (1). We will revise section 4.2.1 to clarify these points.
> -	**Backpropagation through layers:** The backpropagation occurring through the layers only is due to equation (12). Based on the principles of three-factor learning rules, we separate the eligibility traces ($e_{ij}$) from the learning signal in equation (12). Thus, the implementation ensures that the learning signal is computed solely with information from the current time step (i.e., through the layers) by detaching all variables from the previous time step from the differentiable graph.
> -	**Comparison of MACs and memory:** We acknowledge that comparing MACs and memory usage with other works reported in Table 3 may not be entirely fair due to differences in SNN architectures and the focus of other works on architectural and neural model innovations. Therefore, we did not report MACs or memory for such works. Instead, we provided a baseline with BPTT, which serves as the most direct and fair comparison for our experiments, given that the architecture, neuron models, etc., are the same as those used for our method.
>
> [1] Wulfram Gerstner, Marco Lehmann, Vasiliki Liakoni, Dane Corneil, and Johanni Brea. Eligibility Traces and Plasticity on Behavioral Time Scales: Experimental Support of NeoHebbian Three-Factor Learning Rules. Frontiers in Neural Circuits, 12, 7 2018.

---

### Review · Reviewer_7JMW · 2024-08-27

**Summary Of Contributions:**

This paper introduces S-TLLR (STDP-inspired Temporal Local Learning Rule), a mechanism to train deep Spike Neural Networks (SNNs) that takes into account pre- (causal) and post- (non-causal) synaptic activity during training. The main advantage of S-TLLR against competing SNNs training strategies is the favorable memory complexity, $O(n)$, which scales linearly with the number of neurons $n$ and is constant in time $T$. The authors provide estimates of number of MAC operations and memory consumption, as well as accuracy/error performance on a collection of tasks for image, gesture, audio classification and optical flow estimation.

**Audience:**

Yes

**Broader Impact Concerns:**

No concerns one my end.

**Claims And Evidence:**

No

**Requested Changes:**

As outlined in the previous section.

**Strengths And Weaknesses:**

Strengths
- the topic of efficient SNNs training is of interest to the audience of this journal
- the paper is well structured and overall well written, although there's some redundancy in how some concepts (like the computational cost of competing algorithms like backpropagation through time, BPTT) are repeated over and over
- related work is appropriately cited
- estimates on number of MAC operations and memory consumption compare favorably against the BPTT baseline, at the expense of limited accuracy degradation. Hence, S-TLLR could find practical application in resource-constraint environments

Weaknesses
- the main novel contribution of this paper resides in a modification of the eligibility trace formulation to incorporate a secondary activation function in the causal and non-causal terms of eq 10. The authors several choices for activation functions in appendix A.3 and select different functions for different tasks/datasets. It's unclear (and not discussed) what drove this particular selection. There is a limited comparison of outcome in appendix C.3 but how would one go about selecting an activation function for a different task than those described?
- similarly, the choice of hyperparameters ($T_l$, $\lambda_{post}$, $\lambda_{pre}$, $\cdots$) appears to be strongly task-dependent. What kind of search was performed on the hyperparameter space by the authors? How are performance impacted by the choice of hyperparameters?
- Memory consumption and MAC ops are theoretical estimates that don't take into account overheads and constraints associated with real hardware. The authors provide one example of actual measurements on GPU in appendix B.3, demonstrating constant memory consumption of S-TLLR with time $T$. Is the measured ratio between BPTT and S-TLLR memory consumption consistent with the estimated one ($T/2$)?

Typos
- Section 2.2: the term tr$x_j$ is sometimes not set to italics. Please also consider adding parenthesis to the trace function tr($x_j$)
- Section 2.3: respectivetly $\rightarrow$ respectively

---

> ### Author Response · Authors · 2024-08-29
>
> We would like to thank the reviewer for the constructive feedback.
> - **Regarding the role of $\Psi$:** We would like to clarify that the secondary activation function $\Psi$ serves a similar purpose to the surrogate gradient functions used in BPPT (an approximation of the gradient of the firing function in SNNs). As with surrogate gradient functions, different $\Psi$ functions are better suited to particular tasks. We selected four functions that have been used in previous works as surrogate gradient functions and applied them to the tasks where they have demonstrated superior performance. We have provided ablation studies in Table 5 of the Supplemental Material to further validate the choice of functions for specific tasks. We did not include the function shown in Eq. (28) in these studies because it has only been applied in works related to optical flow, and we did not explore it for other tasks. Similarly, we did not conduct ablation studies on other functions for the Optical Flow task due to its computational intensity. Overall, while the secondary activation function enhances the expressiveness of the learning rule, the specific choice of function is a hyperparameter design decision.
> - **Regarding the hyperparameter choices:** As noted by the reviewer, the selection of STDP parameters ($\lambda_{post}$, $\lambda_{pre}$, ⋯) is task-dependent and also influenced by model parameters. For example, $\lambda_{pre}$ controls the exponential decay of causal signals in the learning rule, analogous to the exponential decay in BPTT, as shown in Fig. 1, which corresponds with the leak factor ($\gamma$) of LIF models. Therefore, setting $\lambda_{pre}=\gamma$ is a reasonable choice. However, as demonstrated in the experiment in Section C.2 of the Supplemental Material, setting $\lambda_{pre}$ slightly higher than $\gamma$ can improve model performance. Unfortunately, there is no similar analog parameter for $\lambda_{post}$, so we tested several values and selected the one that yielded the best results. Essentially, these hyperparameters control the temporal dynamics of the learning rule and can be fine-tuned for better performance. However, we did not explore the full hyperparameter space beyond the experiments in Section C.2. Future work could investigate the effects of these parameters further, potentially using meta-learning techniques, as different layers might require different parameters. Regarding $T_l$, this parameter controls the number of updates the model receives, and as shown in the manuscript, more updates result in better performance (e.g., $T_l=0$ outperforms $T_l=5$). However, this also increases the number of MAC operations, making the selection of $T_l$ a trade-off between performance and computational efficiency.
> - **Regarding the MAC and memory estimations:** We acknowledge that the reported values are theoretical estimates that do not consider hardware overheads. We chose to present these theoretical improvements for fairness in comparison, as the values are derived from the algorithms themselves without bias from specific hardware. For example, a GPU might be better suited for BP approaches, while neuromorphic hardware like Loihi might be more suitable for S-TLLR. However, the actual hardware implementation of these algorithms is beyond the scope of this manuscript. Instead, to demonstrate the temporal locality of S-TLLR and highlight the high memory demands of BPTT, we reported memory usage measured on a GPU for both algorithms, as shown in Figure 3. The results in Figure 3 do not align precisely with the theoretical improvement of $T/2$. This discrepancy arises because our code is entirely written in Python using PyTorch. While the functionality is consistent with the described algorithm, memory allocation may not be fully optimized, potentially leading to more memory being allocated than necessary, in contrast to BPTT, which utilizes many well-optimized subroutines. The theoretical results suggest that there is room for improvement in our code implementation. However, it is important to note that fully optimizing the code is hardware-dependent and not the primary goal of this manuscript.
> - **Regarding typos:** We apologize for the typos and have updated the manuscript to correct them.

---

> > ### Comment · Reviewer_7JMW · 2024-08-29
> > **clarification**
> >
> > I appreciated the clarifications and additional information provided.
> >
> > Could you further clarify the exact novelty components presented in this paper? My understanding is that the novelty resides in the combined modifications of the learning rule, such that the eligibility trace:
> > 1. is non-recurrent (setting $\beta = 0$)
> > 2. is modulated by a custom activation function $\psi$ (to some extent analogous to a surrogate gradient function in BPTT)
> > 3. accounts for both causal and non-causal terms
> >
> > Would that be a fair assessment on my part?

---

> > > ### Author Response · Authors · 2024-08-30
> > >
> > > We appreciate the reviewer's insights and would like to emphasize that the primary novelty of our paper lies in the introduction of a new three-factor learning rule, S-TLLR. This rule is inspired by the STDP mechanism and uniquely accounts for non-causal terms in the computation of the eligibility trace. A key advantage of S-TLLR is its memory complexity, which is proportional to the number of neurons and independent of sequence length. This characteristic enhances its practicality for memory-constrained applications, such as on-device adaptation in edge devices.
> > >
> > > Regarding the specific features of the learning rule, we concur with the reviewer's assessment. The non-recurrent eligibility trace computation, the introduction of a custom secondary activation function to improve the expressivity of learning dynamics, and the consideration of both causal and non-causal relationships in spiking activity are indeed central to S-TLLR.

---

### Decision · Action_Editor_jXPY · 2024-12-12

**Recommendation:** Accept with minor revision

**Comment:**

The paper can provide grounds for further research for a not-that-popular research direction in the community. As the claims are also demonstrated (and considering the requested clarifications to be included) and the novelty is sufficient regarding the criteria of TMLR, I suggest acceptance of the paper after a minor revision.

**Audience:**

The paper is interesting to at least the subcommunity of TMLR working on spiking neural networks.

**Claims And Evidence:**

The main claims of the paper regarding the proposed S-TLLR are high accuracy while simultaneously reducing memory and MAC operations in comparison to BPTT.  These results are sufficiently well demonstrated but should be made more concrete in the abstract (i.e., clarify #MAC increases for some settings; clarify "high accuracy" for instance by stating the relative change). Furthermore, as also noted by reviewer 7JMW, it should be clarified that the theoretical insights and the experimental ones are not perfectly aligned (the authors explain this in a comprehensible way but this should be made more clear in relevant parts of the paper).
Overall, I think the claims are sufficiently demonstrated but require the above mentioned clarifications.